# Enhancing Immunogenicity of a Thermostable, Efficacious SARS-CoV-2 Vaccine Formulation through Oligomerization and Adjuvant Choice

**DOI:** 10.3390/pharmaceutics15122759

**Published:** 2023-12-12

**Authors:** Mohammad Suhail Khan, Virginie Jakob, Randhir Singh, Raju S. Rajmani, Sahil Kumar, Céline Lemoine, Harry Kleanthous, Rajesh P. Ringe, Patrice M. Dubois, Raghavan Varadarajan

**Affiliations:** 1Molecular Biophysics Unit (MBU), Indian Institute of Science, Bengaluru 560012, India; mohammadkhan@iisc.ac.in (M.S.K.); rsrajmani@iisc.ac.in (R.S.R.); 2Vaccine Formulation Institute, Rue du Champ-Blanchot 4, 1228 Plan-les-Ouates, Switzerland; virginie.jakob@vformulation.org (V.J.); celine.lemoine@vformulation.org (C.L.); 3Mynvax Private Limited, 3rd Floor, Brigade MLR Centre, No. 50, Vani Vilas Road, Basavanagudi, Bengaluru 560004, India; randhir.singh@mynvax.com; 4Virology Unit, Institute of Microbial Technology, Council of Scientific and Industrial Research (CSIR), Chandigarh 160036, India; sahilkoundal0999@gmail.com (S.K.); rajeshringe@imtech.res.in (R.P.R.); 5Bill and Melinda Gates Foundation, Seattle, WA 98109, USA; hkleanthous@sk.com

**Keywords:** humoral, protein stability, oligomer, COVID-19, protection

## Abstract

Currently deployed SARS-CoV-2 vaccines all require storage at refrigerated or sub-zero temperatures. We demonstrate that after month-long incubation at 37 °C, solubilization, and formulation with squalene-in-water emulsion adjuvant, a stabilized receptor binding domain retains immunogenicity and protective efficacy. We also examine the effects of trimerization of the stabilized RBD, as well as of additional adjuvants, on both B and T-cell responses. The additional emulsion or liposome-based adjuvants contained a synthetic TLR-4 ligand and/or the saponin QS-21. Trimerization enhanced immunogenicity, with significant antibody titers detectable after a single immunization. Saponin-containing adjuvants elicited enhanced immunogenicity relative to both emulsion and aluminum hydroxide adjuvanted formulations lacking these immunostimulants. Trimeric RBD formulated with liposomal based adjuvant containing both TLR-4 ligand and saponin elicited a strongly Th1 biased response, with ~10-fold higher neutralization titers than the corresponding aluminum hydroxide adjuvanted formulation. The SARS-CoV-2 virus is now endemic in humans, and it is likely that periodic updating of vaccine formulations in response to viral evolution will continue to be required to protect vulnerable individuals. In this context, it is desirable to have efficacious, thermostable vaccine formulations to facilitate widespread vaccine coverage, including in low- and middle-income countries, where global access rights to clinically de-risked adjuvants will be important moving forward.

## 1. Introduction

The COVID-19 vaccination campaign is one of the largest in history, with more than 12.7 billion doses administered to people across 184 countries. In the US alone, over 613 million doses have been administered to date. The spike glycoprotein is the most abundant surface glycoprotein on the virus and is extensively utilized in the majority of COVID-19 vaccines currently approved for use or in different stages of clinical development [1,2,3,4,5]. The spike glycoprotein trimer is composed of the surface-exposed S1 and the membrane-anchored S2 subunits. The S1 subunit consists of the N-terminal domain (NTD), the receptor-binding domain (RBD), and two short domains (SD1 and SD2). The RBD binds to the host cell receptor, human angiotensin-converting enzyme 2 (hACE2), and facilitates entry into the cell [6,7]. The majority of neutralizing antibodies generated after SAR-CoV-2 infection or vaccination target the RBD region [8,9]. While several efficacious vaccines for SARS-CoV-2 have been developed and deployed, they all require storage at refrigerated or sub-zero temperatures. This is a significant barrier to widespread deployment in LMICs.

In our previously published study, we introduced three mutations in the RBD to achieve a 7 °C increase in thermal stability compared to the wild-type RBD [10]. The stabilized monomeric RBD, hereafter referred to as R_M, has been demonstrated to elicit ~100-fold higher pseudo-viral neutralizing titers than the WT RBD [10] and, upon lyophilization, retained antigenicity after month-long storage at 37 °C [11]. However, the immunogenicity of the formulation after storage under these conditions has not been examined. To facilitate subsequent clinical development, a CHO cell line expressing R_M was constructed.

Multimerization of antigens interacting with B-cell receptors facilitates the production of high-affinity antibodies compared to monomeric antigens. In another previously published study, we demonstrated that a trimeric WT RBD with a trimerization domain from cartilage matrix protein induces higher antibody titers than monomeric WT RBD [11,12,13]. Adjuvants also play a significant role in modulating both humoral and T-cell responses [14,15]. They are known to enhance antibody titers and provide durable protection. Adjuvants can stimulate an innate response through cellular pattern recognition receptors (PRRs) and activation of signal transduction pathways in antigen-presenting cells such as dendritic cells. Unfortunately, there are very few adjuvants available in the public domain that have been clinically tested in humans. The Vaccine Formulation Institute has recently developed SWE (a squalene-in-water emulsion) which has a similar composition to MF-59^®^, an adjuvant that has a long record of safe use in human influenza vaccines. SWE has been manufactured at GMP grade for use in humans and is now commercialized under the name Sepivac SWE™.

In the present study we compared the immunogenicity and protective efficacy of lyophilized, genetically stabilized monomeric RBD (R_M) stored at 37 °C for one month with that of the same protein stored frozen at −80 °C for the same period. Both proteins were formulated with SWE prior to immunization. The lyophilized protein showed superior immunogenicity and protective efficacy against heterologous viral challenge in hamsters. We also compared the effects of different adjuvants on monomeric (R_M) and trimeric (R_T) stabilized RBD. The adjuvants used were aluminum hydroxide (Al(OH)_3_), SWE, SQ (a squalene-in-water emulsion containing QS21 saponin), SMQ (a squalene-in-water emulsion containing QS21 and a synthetic toll-like receptor 4 (TLR4) agonist 3D(6acyl)-PHAD (3D6AP)), LQ (neutral liposomes containing cholesterol, 1.2-dioleoyl-sn-glycero-3-phosphocholine (DOPC) and QS21), and LMQ (neutral liposomes containing cholesterol, DOPC, QS21, and 3D6AP) [16].

Saponin QS-21, when formulated with cholesterol containing liposomes, has been shown to enhance the uptake of antigens by human monocyte dendritic cells through a receptor-independent, cholesterol-dependent mechanism. In a dose-dependent manner, QS-21 enhances the production of inflammatory cytokines IL-6 and TNF-α [17,18]. QS-21 is also known to trigger caspase-1-dependent release of IL-1β and IL-18 in antigen-presenting cells. While the mechanism of action of QS-21 is not fully understood, some studies suggest that it interacts with lectins through its carbohydrate units, aiding antigen uptake by antigen-presenting cells. Additionally, it also binds to amino groups on T-cell CD2 receptors via imine formation through its triterpene aldehyde, delivering a co-stimulatory signal necessary for T-cell activation and TH1 cellular immunity [19].

All adjuvant formulations induced elevated antibody titers, with the greatest effect observed in QS21-containing formulations. Trimeric stabilized RBD (R_T) induced higher titers of antigen-specific total Ig, IgG2b, and IgG2c than monomeric stabilized RBD. The data collectively indicate that vaccination with either monomeric or trimeric antigens resulted in robust CD4 + T-cell responses dominated by type 1 cytokine expression, along with detectable antigen specific CD8 + T-cell responses

## 2. Material and Methods

### 2.1. Protein Expression and Purification

The plasmids encoding the genes of interest, R_M (stabilized RBD monomer) and, R_T (stabilized RBD trimer), were transfected into Expi293F suspension cells using an ExpiFectamine 293 Transfection Kit (Gibco, Thermo Fisher, Abingdon, UK, Cat # A14524), following the manufacturer’s instructions. Briefly, viable and active cells were passaged at a density of 2 million cells/mL 1 d before the transfection experiment. These cells were then incubated in a humidified 8% CO2 tissue culture shaker incubator at 37 °C (New Brunswick S41i, Edison, NJ, USA). After 1 d, 4 million cells/mL (100 mL) were transiently transfected with the ExpiFectamine–plasmid–DNA complex (100 μg plasmid, complexed with 270 µL of ExpiFectamine 293 reagent). After 20 h, Enhancer 1 and Enhancer 2 were added according to the manufacturer’s protocol, and the mixture was incubated for an additional 4 days for protein production.

A cell-free media supernatant was utilized for the purification of a 6× histidine-tagged protein using immobilized metal affinity chromatography with Ni-NTA resin (G Biosciences, St Louis, MI, USA, Cat # 786940). The cell-free media supernatant was mixed with Ni-NTA resin (5 mL) equilibrated in1× PBS (pH 7.4), and the mixture was incubated for 4 h at 4 °C to allow for protein immobilization under gentle rotation conditions. Subsequently, the complex of cell-free media and resin was gently applied to a column. After collecting the unbound fraction, a wash with 1× PBS (pH 7.4), supplemented with 25 mM imidazole, was performed using ten-column volumes. Bound protein was then eluted using a step gradient of 300 to 500 mM imidazole in 1× PBS (pH 7.4). The eluted fractions were initially analyzed through SDS-PAGE, and fractions containing pure protein were pooled and subjected to dialysis against 1× PBS buffer (pH 7.4) twice, using cellulose dialysis tubing membrane (average flat width 43 mm (1.7 in.), 14 kDa MWCO, Sigma-Aldrich, Singapore, Cat # D9527).

The protein concentration was determined by measuring absorbance at A280 using a BioPhotometer D30 (Eppendorf, Hamburg, Germany), with the theoretical molar extinction coefficients R_M (33,850 M^−1^ cm^−1^) and R_T (37,985 M^−1^ cm^−1^) concentration measured in monomer calculated using the ProtParam tool (https://web.expasy.org/protparam/, accessed on 30 November 2023) (ExPASy, Lausanne, Switzerland). The final dialyzed protein was then analyzed using a 12% SDS-PAGE gel to assess purity and homogeneity under both reducing and non-reducing conditions.

### 2.2. NanoDSF Thermal Melt Studies

Thermal unfolding of trimeric RBD (R_T) was conducted using nanoDSF (Prometheus NT.48) as described by [10]. Experimental measurements were conducted at 100% LED intensity with an initial discovery scan, and the scan counts (350 nm) ranged between 2000 and 3000. Whenever lyophilized protein was utilized, it was reconstituted in water before the DSF experiment.

### 2.3. Size Exclusion Chromatography (SEC)

A Superose 6 Increase 10/300 analytical column (GE Healthcare, Chicago, IL, USA, Code #29091596, Lot #10292150), equilibrated with 1× PBS buffer (pH 7.4), was employed for size exclusion chromatography using the ӒKTA pure™ chromatography system (Cytiva, Marlborough, MA, USA). Approximately 100 µL of purified protein samples (~100 μg) or 5 µL of gel filtration standard, diluted in 100 µL of PBS buffer (BIO-RAD, Hercules, CA, USA; Cat #1511901), was injected into the column at a flow rate of 0.75 mL/min. The peak fraction volume was determined in the Evaluation platform using the peak integration tool (https://www.peakintegration.net/, accessed on 30 November 2023). The molecular weights of these proteins were calculated based on the Protein Standard Log-Molecular Weight Versus Volume plot in GraphPad Prism software (version 9.4.0).

### 2.4. ELISA for Measurement of Serum Binding Antibody Endpoint Titers in hAce2 Expressing Transgenic Mice

Ninety-six-well ELISA plate wells were coated with monomeric RBD (R_M) or spike-2P antigen (4 µg/mL, 50 µL/well, 1× PBS) and incubated at 25 °C under constant shaking (300 rpm) for 2 h using a ThermoMixer (Eppendorf) [13]. Each coated well of the plate was washed three times with 1× PBST (1× PBS with 0.05% Tween-20) (200 µL/well). Each well was then blocked with blocking solution (100 µL, 3% skimmed milk in 1× PBST) and incubated at 25 °C for 1 h at 300 rpm.

In the next step, antisera were serially diluted fourfold, starting from a 1:100 dilution in the blocking solution, and incubated at 25 °C for 1 h at 300 rpm. After sera binding, each well was washed three times (200 µL of 1× PBST/well). Subsequently, Anti-Mouse IgG (Fc specific)–Alkaline Phosphatase antibody produced in goat (Cat # A2429-Sigma-Aldrich, Lot# 029M4801V), diluted 1:5000 in blocking buffer (50 µL/well), was added and the mixture was incubated at 25 °C for 1 h at 300 rpm.

After incubation, three washes were performed (200 µL of 1× PBST/well). Finally, each well was incubated with Alkaline Phosphatase Yellow (pNPP) Liquid Substrate (50 µL/well; Cat# P7998, Sigma-Aldrich) at 37 °C for 30 min. The chromogenic signal was measured at 405 nm. The highest serum dilution with a signal above the cut-off (0.2 O.D. at 405 nm) was considered the endpoint titer for the ELISA.

### 2.5. SARS-CoV-2 Pseudovirus Preparation and Neutralization Assay

HIV-1-based pseudotyped viruses were employed in pseudoviral neutralization assays, following a method previously described [11]. Briefly, adherent HEK293T cells were transiently transfected with plasmid DNA pHIV-1 NL4•3Δenv-Luc and Spike-Δ19-D614G, using the ProFection mammalian transfection kit (Cat# E1200, Promega Inc., Singapore) for pseudovirus production. The culture supernatant was harvested 48 h post-transfection, filtered through a 0.22 μm filter, and stored at −80 °C. Adherent HEK293 cells expressing hACE-2 and TMPRSS2 receptors (BEI resources, NIH, Catalog No. NR-55293) were cultured in a growth medium consisting of DMEM with 5% Fetal Bovine Serum (Thermo Fisher) and penicillin–streptomycin (100 U/mL). Mice serum samples were heat-inactivated and then serially diluted in the growth medium, starting from 1:20 dilutions. In the next step, the pseudotyped virus was incubated with the serially diluted sera in a total volume of 100 µL for 1 h at 37 °C. The adherent cells were then trypsinized, and 1 × 10^4^ cells/well were added to achieve a final volume of 200 µL/well. The plates were further incubated for 48 h in a humidified CO_2_ incubator at 37 °C. After incubation, neutralization was measured as an indicator of luciferase activity in the cells (relative luminescence units) using Nano-Glo luciferase substrate (Cat # N1110, Promega). Luminescence was measured using a Cytation-5 multimode reader (Bio-Tech Inc., Oklahoma City, OK, USA). The luciferase activity, measured as relative luminescence units (RLU), in the absence of sera was considered as 100% infection. The serum dilution resulting in half-maximal neutralization of the pseudovirus (ID_50_) relative to the no-serum control was determined from neutralization curves.

### 2.6. Adjuvant Preparation and Formulation Characterization

Aluminum hydroxide gel (Alhydrogel 2%) was purchased from Croda. The SWE adjuvant (squalene-in-water emulsion) was co-developed by the Vaccine Formulation Institute (Plan-les-Ouates, Switzerland) and Seppic (La Garenne Colombes, France) and is available at GMP grade (Sepivac SWE™) under an open-access model. The SQ, SMQ, LQ, and LMQ adjuvant formulations were manufactured at the Vaccine Formulation Institute (VFI) [16].

The SQ adjuvant was prepared by combining a solution of QS21 saponin (Desert King International, San Diego, CA, USA) with a squalene-in-water emulsion containing cholesterol. The SMQ adjuvant was prepared similarly, but with the incorporation of a synthetic TLR4 agonist, 3D(6acyl)-PHAD (3D6AP) (Avanti Polar Lipids, Alabaster, AL, USA). The LQ adjuvant was prepared by adding a QS21 solution to neutral liposomes composed of dioleoyl phosphatidylcholine (DOPC) and cholesterol. The LMQ adjuvant is based on LQ with the inclusion of 3D6AP. Injectable formulations were prepared with LQ, SQ, LMQ, and SMQ at 2 μg of the TLR4 agonist 3D6AP and/or 5 μg of QS21 saponin per injected dose. The SWE, SQ, and SMQ emulsion adjuvants contained 1 mg of squalene per injected dose. The compatibility of R_M and R_T antigens with the panel of adjuvants was assessed through a comprehensive physicochemical characterization of each adjuvant in formulations prepared at a 1:1 volume ratio and stored for 24 h at 5 °C. Integrity of the R_M and R_T antigens in adjuvanted formulations was monitored using a sandwich ELISA that was essentially as previously described [17]. Briefly, R_M and R_T adjuvanted formulations were added to 96-well plates coated with human mAb CR3022 anti-SARS-CoV-2 RBD (Abcam, Cambridge, UK) and bound antigen was detected with a biotinylated human ACE2-Fc fusion protein (Institute of Protein Design, Seattle, WA, USA).

### 2.7. Mouse Immunization for Characterization of B- and T-Cell Responses

Female C57BL/6J mice aged 6 to 8 wk (Charles River) were intramuscularly immunized in the hind leg gastrocnemius muscle on days 0 and 21 with 50 µL of either adjuvanted vaccine or control formulations (excipient alone or antigen + excipient). To track antibody responses, blood was collected from mice (n = 6) on days 20 and 42. The collected blood was centrifuged at 10,000× *g* for 10 min, and the serum layer was carefully collected and stored at −80 °C for subsequent analysis. To monitor T-cell responses, spleens were harvested from mice (n = 4) on day 28 and processed for intracellular cytokine staining (ICS) as described previously [17].

### 2.8. ICS Method for T-Cell Responses (IL-2, IFNγ and Th2 Cytokines IL-4, IL-5 and IL-13)

Splenocytes (1 × 10^6^) were stimulated in 96-well plates with a pool of 20mer peptides with 10aa overlap, covering the entire RBD sequence (Genscript, Piscataway, NJ, USA) at 1.5 µg/mL/peptide or medium, both containing 1 µg/mL anti-mouse αCD28 (BD Pharmingen, San Diego, CA, USA) and incubated for 2 h at 37 °C in round-bottom 96-well plates. Cytokine secretion was then blocked using BD GolgiPlug™ protein transport inhibitor and cells were further incubated for 16 h at 37 °C and 5% CO_2_. Cells were then placed on ice, transferred to V-bottom 96-well plates, and stained for 15 min at 4 °C using a LIVE/DEAD™ cell stain (Life Technologies, Austin, TX, USA). Cell surface staining was performed for 15 min on ice with anti-CD3e (clone 17A2)-PE/Cy7 and anti-CD8a (clone 53-6.7)-BV605 at 1:400 dilution, anti-CD4 (clone GK 1.5)-PerCP/Cy5.5 at 1:800 (BioLegend^®^, San Diego, CA, USA), and CD16/32-Fc Block (2.4G2) (BD Pharmingen) at 1:100 dilution. After washing, fixation, and permeabilization using Cytofix/Cytoperm™ (BD Biosciences, Alpharetta, GA, USA), cells were incubated at 1:200 dilution for intracellular staining for 30 min at 4 °C with anti-IL2-(clone JES6-5H4)-AF488, anti-IL5 (clone TRFK5)-BV421, anti-IL4 (clone 11B11)-BV421 (BioLegend), anti-IL13 (clone eBio13A)-efluor450 (Invitrogen, Waltham, MA, USA), and anti-IFNγ-PE (BD Biosciences). Cells were washed twice with Perm/Wash buffer (BD Biosciences) and resuspended in 300 µL PBS. FACS analysis was completed with an Attune™ NxT equipped with a Cytkick auto-sampler. Compensation matrices were confirmed using an ArC™ Amine Reactive Compensation Bead Kit (Invitrogen). FACS data were analyzed using FlowJo software (version 10.1 )(FlowJo LLC, Vancouver, BC, Canada).

### 2.9. ELISA for Quantitating Antibody Responses in C57Bl/6 Mice

Anti-SARS-CoV-2 Ig responses were measured in mouse sera using enzyme-linked immunosorbent assays (ELISA). Nunc MaxiSorp™ plates (Thermo Fisher Scientific, Waltham, MA, USA) were coated directly with Monomeric soluble RBD (Institute of Protein Design) at 1.25 µg/mL in 1× PBS overnight at 4 °C, and blocked for 1 h with PBS-t, PBS containing 0.05% (*v*/*v*) Tween 20 (Applichem, Boca Raton, FL, USA), and 2% (*w*/*v*) bovine serum albumin (BSA) (Sigma, Tokyo, Japan). The plates were washed with PBS-t and incubated with 4× serial dilutions of mouse serum samples diluted 1/100 in PBS containing 0.05% (*w*/*v*) BSA. Plates were incubated for 1 h at 37 °C, washed, and then incubated for 1 h at 37 °C with anti-mouse antibodies coupled to HRP (Southern Biotech, Birmingham, AL, USA) including goat anti-mouse total Ig, IgG1, IgG2b, or IgG2c, an allelic form of IgG2a expressed in C57BL/6 mice [20,21] at 1:6000 dilution. Plates were then washed before the addition of TMB supersensitive (Sigma, T4444) substrate solution. The reaction was stopped at 7 min by adding 1M sulfuric acid. Absorbance was directly measured at 450 nm using a microplate reader (Biotek Instruments, Winooski, VT, USA).

### 2.10. IFNγ Cytokine Quantification by ELISA

Splenocytes were incubated with RBD peptides (20mers with 10aa overlap) without a cytokine secretion inhibitor. Supernatants were then collected after 48 h and stored at −80 °C. IFNγ, secretion were quantified in the culture supernatants using ELISA kits according to the manufacturer’s instructions (Thermo Fisher Scientific, Cat. 88-7314-88 [IFNγ]).

### 2.11. Challenge Studies

The animals were moved to the virus BSL-3 laboratory at the Centre for Infectious Disease Research, Indian IISc, (Bangalore, India). They were housed in individually ventilated cages (IVC) at a temperature of 22 ± 1 °C and a relative humidity of 50 ± 5%. After a period of 7 d to acclimate to the IVC cages, the immunized animals were sedated and anesthetized using a combination of ketamine (90 mg/kg for mice) and xylazine (4.5 mg/kg for mice). They were then intranasally exposed to either 10^4^ pfu Beta, 10^4^ pfu Delta, or 7.5 × 10^3^ BA.1 VOCs. The unimmunized animals served as the control group. The weight changes of the immunized-challenged animals, unimmunized-unchallenged animals (unimmunized), and unimmunized-virus challenged control groups were monitored and recorded for a period of 5–9 d. Between 6–10 d post-challenge, all of the animals (mice and hamsters) were euthanized using an intraperitoneal injection of a ketamine and xylazine overdose, followed by cervical dislocation for humane termination in the case of mice. To assess lung tissue histopathology, we utilized a modified version of Mitchison’s virulence scoring system, which takes into account lung consolidation, severity of bronchial and alveolar inflammation, immune cell influx, and alveolar and perivascular edema. The histopathology scores ranged from 0 to 4, with 4 indicating severe pathology and 0 indicating no pathology.

## 3. Results

### 3.1. Protein Expression and Characterization

R_M and R_T were expressed in Expi293 cells by transient transfection and purified using Ni-NTA chromatography as described previously [11]. As observed previously, they formed monomers and disulfide-linked trimers, respectively (Figure 1A,C). We have previously shown that, upon lyophilization, the stabilized monomer was stable to ~90 °C temperatures for one hour and ~one month at 37 °C [10]. Here we show that this holds true for the stabilized trimer as well (Figure 1B,D). To facilitate clinical development, we constructed stable CHO cell lines expressing tag free R_M. In order to examine the immunogenicity of R_M following thermal stress, we incubated CHO expressed, purified and lyophilized R_M at 37 °C for one month. Following this, the protein was resolubilized and characterized by SEC (Figure 2). The lyophilized and thermally stressed protein showed the presence of some heterogeneous aggregates, but the major peak remained monomeric.

### 3.2. Immunization and Challenge Studies with R_M in hACE2 Expressing Transgenic Mice

While we had previously shown that the antigenicity of lyophilized R_M was unaffected by one month-long storage at 37 °C, the immunogenicity of the thermally stressed material remained to be characterized. We chose R_M rather than R_T for these challenge studies, as it was expressed at higher yield and R_T showed low but clearly measurable ELISA titers directed against the trimerization domain [11]. Hence, lyophilized R_M that had been incubated at 37 °C for one month and resolubilized was formulated with SWE. As a control, freshly thawed protein that had been stored frozen in PBS at −80 °C was also similarly formulated. Two intramuscular immunizations were administered three weeks apart, and sera collected two weeks after each immunization. The sera after the second immunization were characterized by ELISA and neutralization titers (Figure 3). Surprisingly, the lyophilized and thermally stressed protein showed enhanced immunogenicity relative to the control. Following prime and boost immunizations, the animals were subjected to heterologous challenge with either Beta or Delta virus (Figure 4 and Figure 5). In both cases, consistent with the enhanced neutralization, the lyophilized and thermally stressed protein was associated with lower weight loss in vaccinated animals. The Omicron BA.1 virus has a large number of mutations compared to the B.1 virus. Therefore, it was of interest to compare protection conferred by stabilized RBD (R_M) derived from B.1 and BA.1 and sequences against a significantly evolved Omicron BA.1 challenge strain. Surprisingly despite the large number of immune evading mutations present in BA.1, R_M derived from B.1 conferred similar protection to R_M derived from BA.1 (Figure 6).

### 3.3. Potency and T Helper Bias of Adjuvanted, Formulated Material

Protection against SARS-CoV-2 infection is primarily achieved through neutralizing antibodies that prevent the virus from binding to host ACE2 receptors. Antibodies can also control virus spread in vivo through antibody-dependent cellular cytotoxicity (ADCC) and other Fc-dependent effector function [22,23,24]. To assess humoral responses elicited by vaccination, C57BL/6 mice were immunized twice on days 0 and 21 with formulations containing monomeric R_M or trimeric R_T immunogens, combined with liposomal- and squalene-based emulsion adjuvants, or aluminum hydroxide. RBD-specific antibody responses were measured in sera collected 21 days after the boost. A full-length prefusion spike trimer antigen, adjuvanted with the oil-in-water emulsion SWE, was used as a comparator. While all groups vaccinated with monomeric RBD (R_M) or trimeric RBD (R_T), with or without adjuvant, exhibited detectable levels of antibodies, adjuvanted formulations induced significantly higher total Ig responses and IgG subclass responses than non-adjuvanted antigens. The R_T antigen consistently generated higher titers of antigen-specific total Ig, IgG2b, and IgG2c than the R_M antigen, with antibodies evident after a single dose. The IgG1 to IgG2c ratio was lower in groups vaccinated with the R_T relative to R_M and, in the former, was lowest in groups with QS21-containing adjuvants (Figure 7).

### 3.4. pVNT Results for R_M and R_T Immunogens with Sera in from C57BL/6 Mice

The results depicted in Figure 8 reveal that all groups vaccinated with adjuvanted antigens elicited significantly elevated pseudoviral neutralization titers (pVNT) against both the WT and Delta strains compared to those observed in the excipient and non-adjuvanted antigen groups. Notably, the trimeric RBD (R_T) antigen elicited substantially higher pseudoviral neutralization titers (pVNT) when compared to the monomeric RBD (R_M) antigen. Neutralizing titers against Omicron were observed only for R_T with SMQ and LQ adjuvants [25,26] (Figure 8).

### 3.5. T-Cell Responses Elicited by R_M and R_T Formulations

Vaccine antigen-specific T-cell responses constitute a vital component of the adaptive immune response, contributing significantly to protection against pathogens. CD4 T-cells play a crucial role in induction of antibody responses by providing co-stimulatory signals to antigen-specific B-cells, amplifying immunoglobulin production, promoting class switching, enhancing the secretion of effector cytokines, and exhibiting cellular cytotoxic activity, collectively contributing to pathogen control [27] Additionally, CD4 T-cells offer essential help to CD8 T-cells by engaging CD40 on antigen-presenting cells [28]. The CD8 T-cell subset contributes to viral load control through cytokine secretion and cytotoxic activity, targeting virally infected cells.

T-cell responses specific to the R_M and R_T antigens were assessed in mice vaccinated on days 0 and 21 with formulations containing various adjuvants. Spleen cells collected seven days after the boost were stimulated with a pool of 20mer peptides overlapping by 10 amino acids, spanning the entire RBD antigen sequence. Our findings indicate that CD4 T-cells and CD8 T-cells expressing type 1 cytokines IL-2 and/or IFN-γ were detectable in all groups vaccinated with formulations containing QS21 (Figure 9, Figure 10, Figure 11 and Figure 12). For both antigens, type 1 CD4 responses were generally lower when AlOH and Sepivac SWE™ adjuvants, which lack the QS21 saponin, were employed. The cumulative expression of type 2 cytokines IL-4, IL-5, and IL-13 remained low (0.4%) across all groups, with the highest responses observed in mice vaccinated with the full-length spike protein adjuvanted with SWE. As with antibody responses, R_T elicited higher T- cell responses than R_M. The AS01 mimic, LMQ was the best at inducing Th1 type responses in both CD4+ and CD8+ T-cells.

## 4. Discussion

Protection against SARS-CoV-2 infection is primarily achieved through neutralizing antibodies that prevent the virus from binding to host ACE2 receptors. Antibodies can also control virus spread in vivo through antibody-dependent cellular cytotoxicity (ADCC) and other effector function-mediated mechanisms [22,23,24]. In the present study we observed that stabilized monomeric RBD (R_M), upon lyophilization, storage at 37 °C for one month and subsequent formulation with SWE, showed slightly but statistically significant enhanced immunogenicity, relative to freshly formulated protein that had been previously stored at −80 °C. We have previously shown [10,11,13] that the protein retains conformation and antigenicity after multiple freeze–thaw cycles. It is possible that the aggregates that are formed after extended storage at 37 °C may contribute to this enhanced immunogenicity and protective efficacy.

Trimerization of RBD results in elicitation of higher antibody titers and improved neutralization responses although sera contain small but measurable titers against the trimerization domain [13,27]. In addition, incorporation of stabilizing mutations significantly improves the immunogenicity of WT-RBD [10]. In the present study, we compared the immunogenicity of stabilized monomer (R_M) and stabilized trimer (R_T) formulated with several adjuvants. R_T containing formulations consistently elicited higher antibody and neutralization titers compared to corresponding R_M formulations. Adjuvanted formulations containing either R_M or R_T elicit enhanced antibody and neutralization titers relative to corresponding formulations without adjuvant, with a notable increase seen in formulations incorporating QS21 adjuvants. The R_T/LMQ formulation elicits the highest neutralization titers, and this most clearly seen for the Omicron BA.1 neutralization data in Figure 8F. LMQ is similar to the AS01 adjuvant; interestingly, the latter was not extensively used during the pandemic. The response of type 2 cytokines IL-4, IL-5, and IL-13 remained low across groups, except in the case of animals immunized with full-length spike adjuvanted with SWE. This may be related to the type of antigen, as monomeric and trimeric RBDs induce lower type 2 cytokine expression in T-cells than the trimeric spike antigen. Alternatively, timing of in vitro cultures and the intracellular cytokine staining assay used in this study may be suboptimal for detection of antigen specific T-cells expressing IL-4, IL-5 and IL-13. In future studies, we will examine whether adjuvants affect both the quantity and quality of immune responses, both neutralizing and non-neutralizing, and examine the duration of the elicited antibody responses.

The observation that lyophilized RBD retains immunogenicity upon extended thermal stress suggests that such lyophilized formulations can be deployed without cold-chain requirements, facilitating use in LMICs. Despite this important result, there are manufacturing complications associated with using and dispensing lyophilized material in single dose formulations. In addition, this modality requires two vial formulations and bedside mixing. In the future we will examine the immunogenicity of adjuvanted RBD formulations such as those described in the present work after subjecting these to varying degrees of thermal stress, with the goal of developing efficacious, adjuvanted, single-vial, liquid COVID-19 vaccine formulations that are stable for at least two weeks at elevated temperatures of up to 40 °C. The SARS-CoV-2 virus is now endemic in humans, and it is likely that periodic updating of vaccine formulation in response to viral evolution will continue to be required to protect vulnerable individuals. Therefore, it is desirable to have efficacious, thermostable vaccine formulations to facilitate widespread vaccine coverage, including in LMICs. In this context, clinical testing of formulations containing globally accessible adjuvants, such as those employed in the present work, will be extremely important, moving forward.

## Figures and Tables

**Figure 1 pharmaceutics-15-02759-f001:**
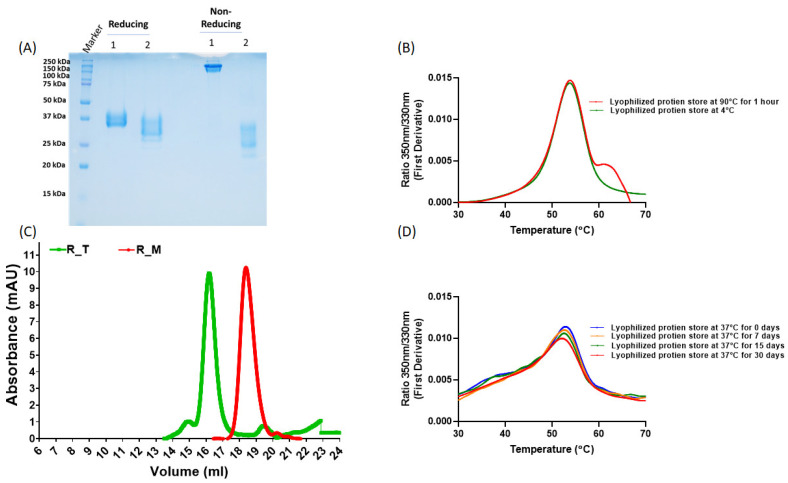
Biophysical characterization of RBD. (**A**) 12% SDS PAGE with (1) R_T and (2) R_M under reducing and non-reducing conditions. (**B**) Lyophilized R_T was subjected to 90 °C incubation for 1 h and, following solubilization, nDSF-data were acquired. Similar data for R_M have been reported previously [10] (**C**) Size-exclusion chromatography profiles for monomeric RBD (R_M) and trimeric RBD (R_T) with predominant peaks at 16.78 mL and 14.01 mL, respectively, on an S200 10/300GL column calibrated with Biorad gel filtration marker (Hercules, CA, USA, Cat. No. 1511901), run at a flow rate of 0.5 mL/min with PBS (pH 7.4) as mobile phase. (**D**) Equilibrium thermal unfolding measured using nanoDSF for lyophilized R_T, subjected to 37 °C incubation for 4 weeks, and then solubilized prior to nDSF.

**Figure 2 pharmaceutics-15-02759-f002:**
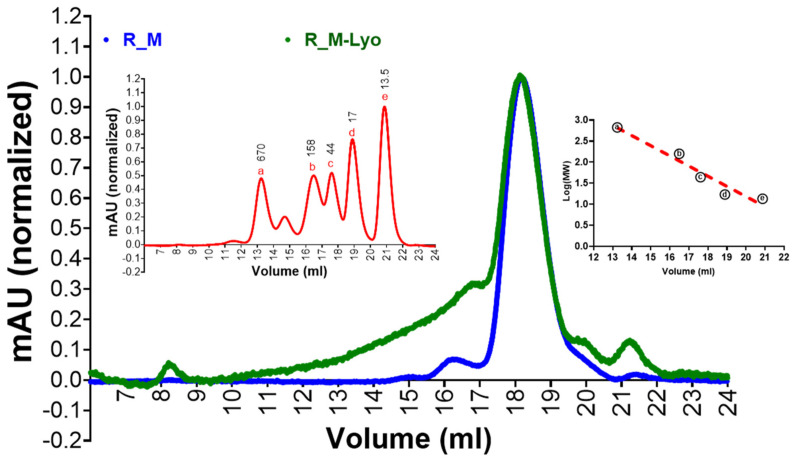
Effect of lyophilization and extended thermal stress on oligomerization status and antigenicity of R_M. Protein in PBS buffer was lyophilized, incubated at 37 °C for 4 weeks and then redissolved in an identical volume of water prior to the experiments. The control sample was stored in PBS at −80 °C and thawed prior to use. Protein samples were subjected to SEC on a Superose 6 10/300 GL prepacked column (Cytiva). Peak maxima normalized plot for proteins is shown in the main panel. SEC profile of Gel Filtration Standard (BioRAD, Hercules, CA, USA, Cat #1511901) is shown in the left inset, whereas a–e corresponds to molecular weights of standards shown in left inset and marked as kDa values (670, 158, 44, 17 and 13.5 respectively while Log MW vs elution volume plot is shown in the right inset. The elution volume for R_M = 18.1632 mL and R_M-Lyo = 18.16952. The corresponding protein molecular weight of R_M = 42.277 kDa and R_M-Lyo = 42.141 kDa.

**Figure 3 pharmaceutics-15-02759-f003:**
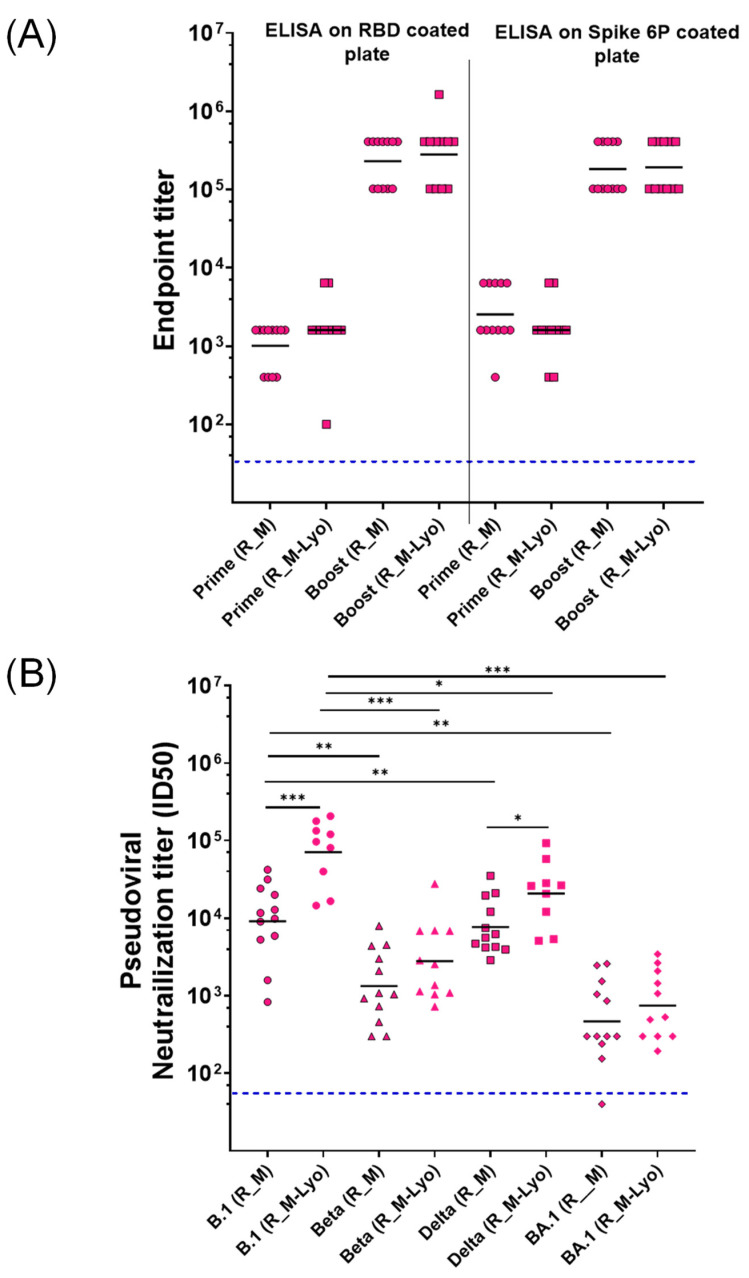
Lyophilized R_M retains immunogenicity after one month-long incubation at 37 °C. ELISA titers elicited by R_M and R_M-Lyo prime and Boost sera on SARS COV-2 RBD and Spike-6P immobilized (4 µg/mL) ELISA plates (**A**). Boost sera were tested for neutralization against SARS CoV-2 B.1 (WT), Beta, Delta, and BA.1 (Omicron) pseudoviruses. Lyophilized and thermally stressed protein elicits sera with superior immunogenicity, compared to the freshly formulated control protein (**B**). Statistical significance was determined on log-transformed data by one-way ANOVA with Dunnett’s multiple comparison (* *p* ≤ 0.05; ** *p* ≤ 0.01; *** *p* ≤ 0.001).

**Figure 4 pharmaceutics-15-02759-f004:**
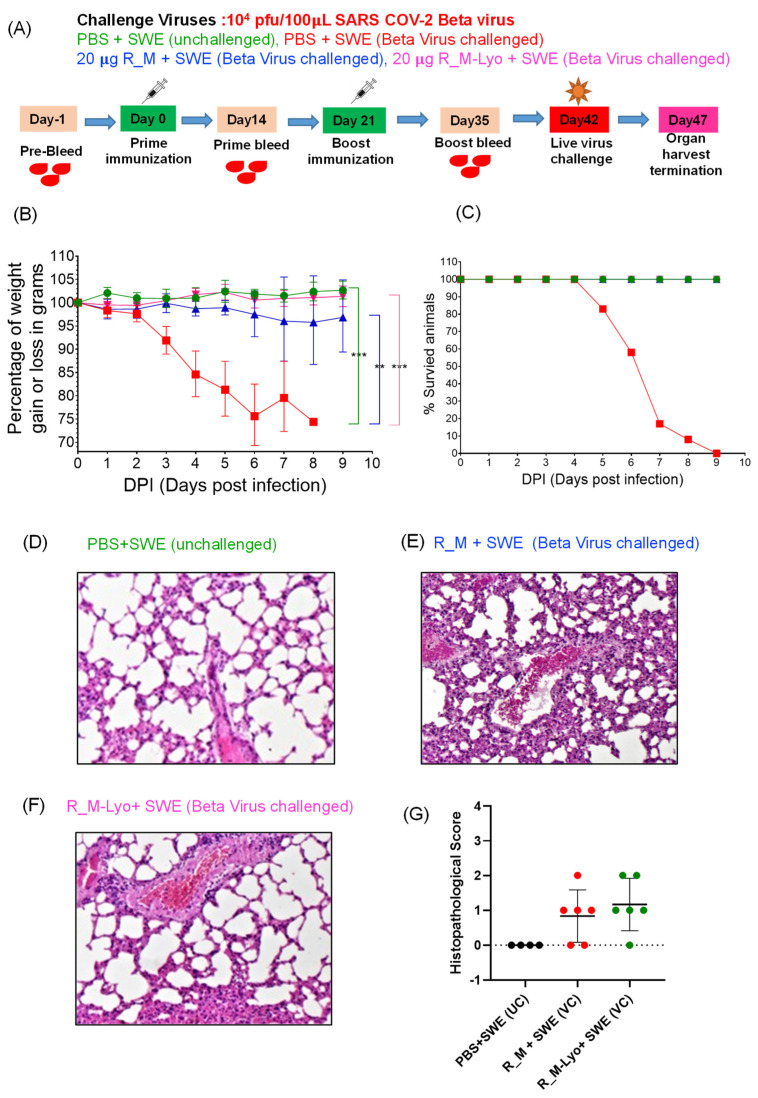
Lyophilized R_M retains protective efficacy against heterologous Beta virus challenge in transgenic mice expressing human ACE2 (hACE-2 mice), after one month-long incubation at 37 °C. (**A**) hACE-2 expressing transgenic mice were immunized twice with 20 µg of SWE adjuvanted R_M control or R_M-Lyo protein. The control protein was stored frozen at −80 °C and thawed on ice prior to formulation with SWE. The lyophilized protein was incubated at 37 °C for one month, reconstituted in water at 1 mg/mL, and then formulated with SWE. Three weeks after the second immunization, mice were subjected to intranasal challenge with 10^4^ pfu of live SARS-CoV-2 Beta variant. (**B**,**C**) Average body weight changes and % survival at nine days post-virus challenge. For statistical comparison, the mean of two independent groups were analyzed using Unpaired *t*-test (** *p* ≤ 0.01; *** *p* ≤ 0.001) with GraphPad Prism 9.4.0. Each condition [green (unchallenged), blue (R_M + SWE Beta Virus challenged) and pink (R_M-Lyo + SWE Beta Virus challenged) were independently compared with PBS+SWE (Beta virus challenge, red color). (**D**–**F**) Lung histopathology images at 10× magnification are shown for all cases except the unimmunized Beta virus challenge group. None of the unvaccinated challenged animals survived beyond 8 days. (**G**) Histopathological score.

**Figure 5 pharmaceutics-15-02759-f005:**
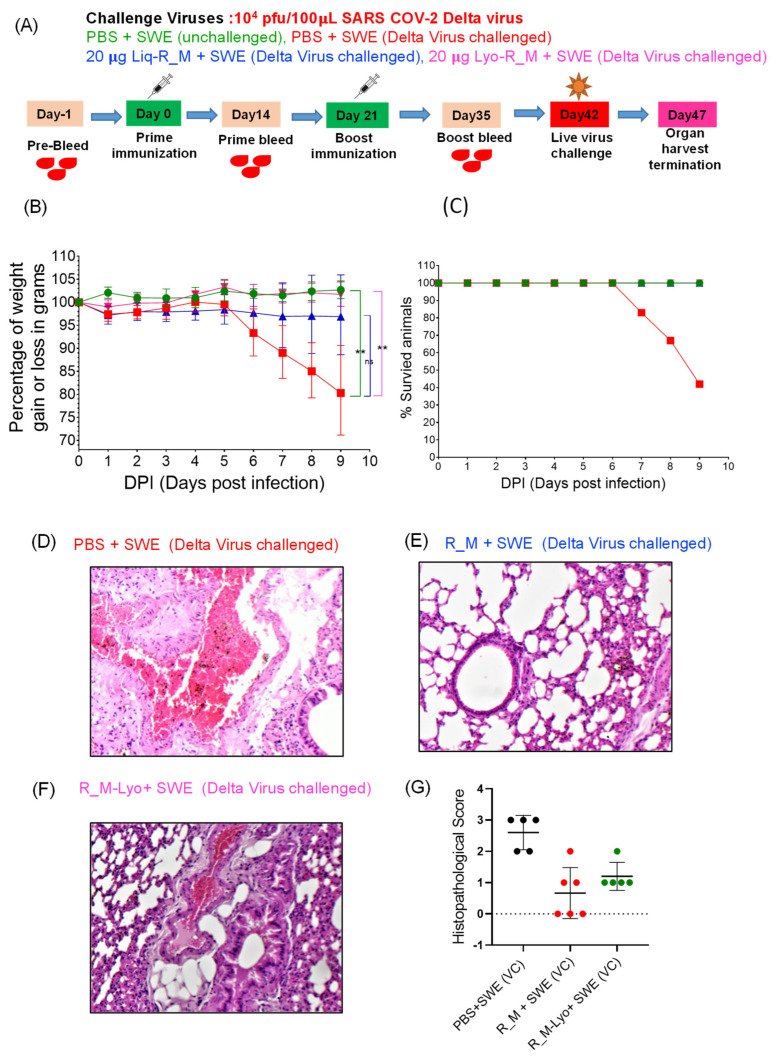
Lyophilized R_M retains protective efficacy against heterologous Delta virus challenge in transgenic hACE-2 mice, after one month-long incubation at 37 °C. hACE-2 expressing transgenic mice were immunized twice with SWE adjuvanted 20 µg of R_M and R_M-Lyo protein. Sub-zero protein R_M was thawed slowly at 4 °C and lyophilized R_M-Lyo protein was first reconstituted in water at 1 mg/mL and then formulated with SWE. Animals were subjected to intranasal challenge with 10^4^ pfu of live SARS-CoV-2 delta variants 21 days post-boost immunization. (**A**) A day-wise immunization scheme with the study group and challenge virus dose is shown in depiction (**B**,**C**) Animal average body weight changes and % animal survival after nine days post-virus challenges are shown in two separate panels, (**B**) and (**C**), respectively. For statistical comparison, the mean of two independent groups were analyzed using Unpaired *t*-test (non-significant “ns” *p* ≥ 0.05; ** *p* ≤ 0.01) with GraphPad Prism 9.4.0. Each condition [green (unchallenged), blue (R_M + SWE delta Virus challenged) and pink (R_M-Lyo + SWE delta Virus challenged) were independently compared with PBS+SWE (delta virus challenge, red color). (**D**–**F**) Lung histopathology image at 10× magnification. None of the unvaccinated challenge animals survived beyond 9 days. (**G**) Histopathological score.

**Figure 6 pharmaceutics-15-02759-f006:**
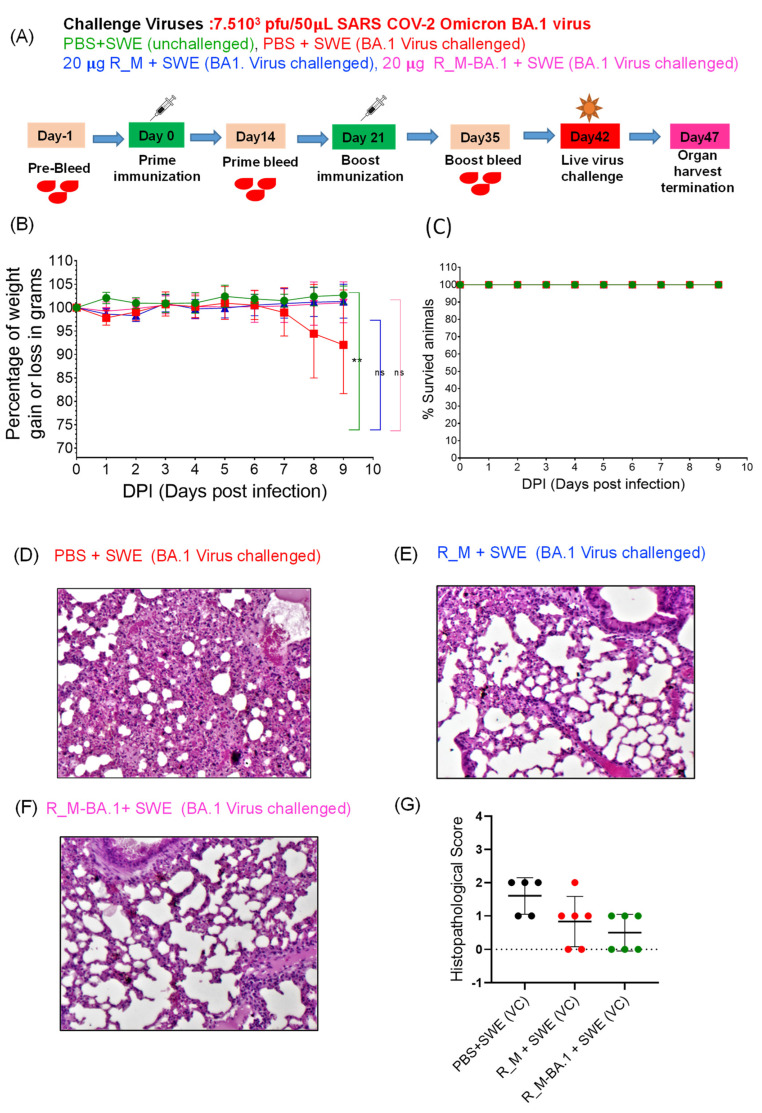
R_M protein protects immunized hACE-2-transgenic mice against heterologous Omicron (BA.1) challenge. hACE-2 expressing transgenic mice were immunized twice with SWE adjuvanted 20 µg of R_M and R_M BA.1 protein. Sub-zero protein R_M and R_M-BA.1 were thawed slowly at 4 °C and then formulated with SWE. Animals were subjected to intranasal challenge with 7.5 × 10^3^ pfu of live SARS-CoV-2 Omicron BA.1 variant 21 days post-boost immunization. (**A**) A day wise immunization scheme with study group and challenge virus dose is shown in depiction. (**B**,**C**) Animal average body weight changes and % animal survival after nine days post-virus challenges are shown in two separate panels, (**B**) and (**C**), respectively. For statistical comparison, the mean of two independent groups were analyzed using Unpaired *t*-test (non-significant “ns” *p* ≥ 0.05; ** *p* ≤ 0.01) with GraphPad Prism 9.4.0. Each condition [green (unchallenged), blue (R_M + SWE BA.1 Virus challenged) and pink (R_M-BA.1 + SWE BA.1 Virus challenged) were independently compared with PBS+SWE (BA.1 virus challenge, red color). (**D**–**F**) Lung histopathology image at 10× magnification. (**G**) Histopathological score.

**Figure 7 pharmaceutics-15-02759-f007:**
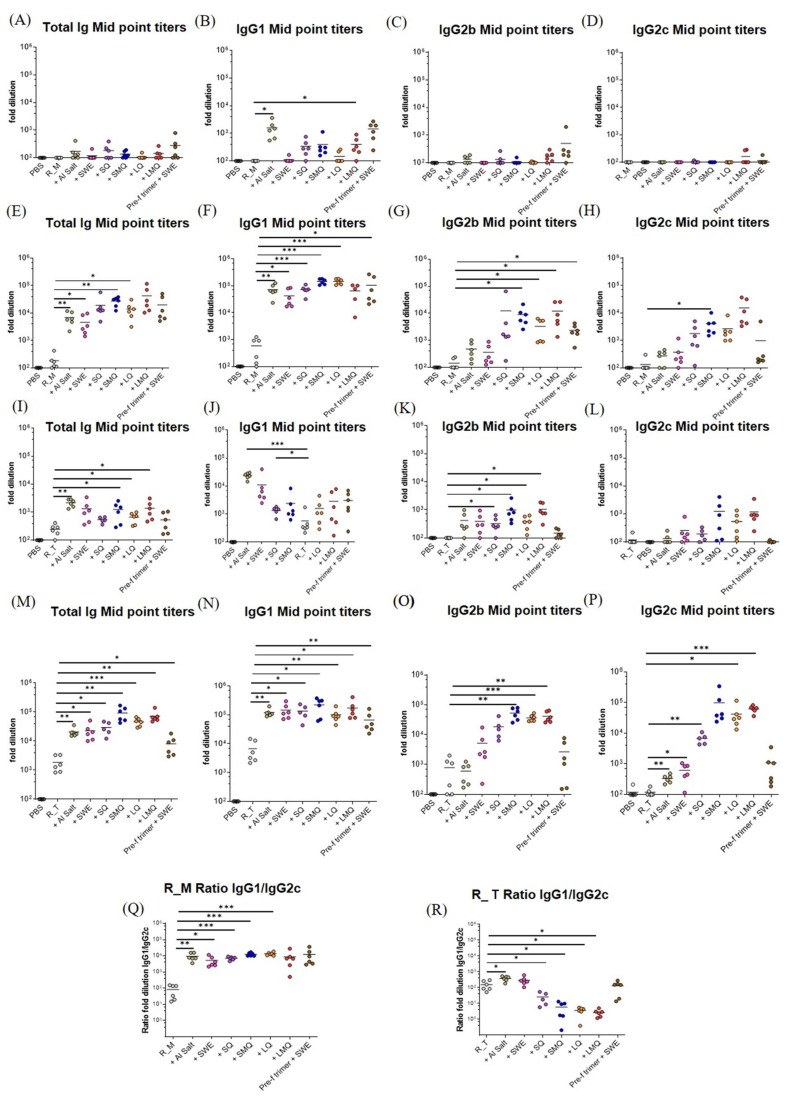
ELISA EC50 titers elicited by R_M and R_T with different adjuvants in C57BL/6 mice. Prime with R_M (monomeric RBD): (**A**) Total Ig midpoint titer, (**B**) IgG1 titer, (**C**) IgG2b titer, (**D**) IgG2c titer. Boost with R_M (monomeric RBD): (**E**) Total Ig titer, (**F**) IgG1 titer, (**G**) IgG2b titer, (**H**) IgG2c titer. Prime with R_T (trimeric RBD): (**I**) Total Ig titer, (**J**) IgG1 titer, (**K**) IgG2b titer, (**L**) IgG2c titer. Boost with R_T (trimeric RBD): (**M**) Total Ig titer, (**N**) IgG1 titer, (**O**) IgG2b titer, (**P**) IgG2c titer. The ratio of IgG1 and IgG2c ELISA titers for (**Q**) R_M and (**R**) R_T. Data are n = 6 mice per group. Statistical significance was determined on log-transformed data by one-way ANOVA with Dunnett’s multiple comparison (* *p* ≤ 0.05; ** *p* ≤ 0.01; *** *p* ≤ 0.001).

**Figure 8 pharmaceutics-15-02759-f008:**
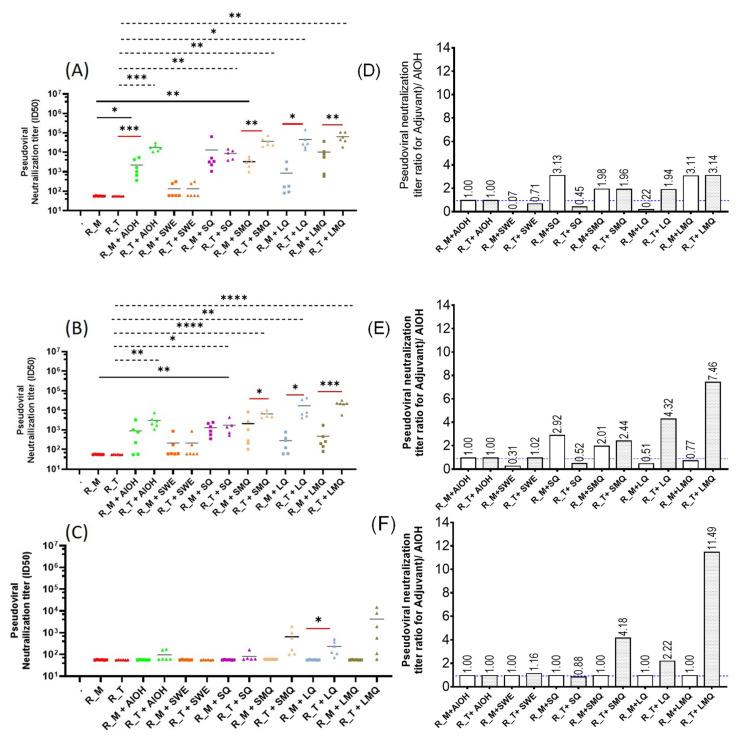
Pseudoviral neutralization titers (pVNT) elicited in C57BL/6 mice. Pseudoviral neutralization titers elicited by the various formulations against (**A**) WT, (**B**) Delta, (**C**) Omicron (BA.1). *p* values for comparisons were calculated with a two-tailed Mann–Whitney test. All sera that showed a titer of <60 were assigned a value of 60 for purposes of comparison. Relative pseudoviral neutralization titer (Titer (Adjuvant X)/) Titer (AlOH)) for the following pseudoviruses: (**D**) WT, (**E**) Delta, (**F**) Omicron (BA1). The geometric mean pVNT ID50 for all six animals in each group has been used in the analyses. Data are presented as individual responses of n = 6 mice with the mean (bar). Statistical significance was determined on log-transformed data by one-way ANOVA with Dunnett’s multiple comparison (* *p* ≤ 0.05; ** *p* ≤ 0.01; *** *p* ≤ 0.001; **** *p* ≤ 0.0001).

**Figure 9 pharmaceutics-15-02759-f009:**
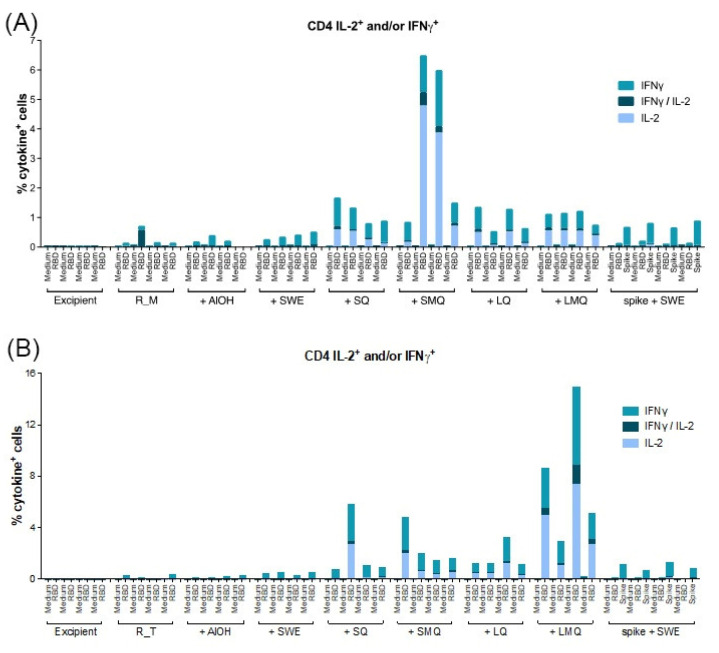
Quantification of CD4 T-cells expressing type 1 cytokines elicited by monomeric RBD (R_M) (top panel) and trimeric RBD (R_T) (bottom panel) in C57BL/6 mice. C57BL/6 mice were immunized IM on Day 0 and Day 21 with R_M (**A**) or R_T (**B**) adjuvanted with AlOH, SWE, SQ, SMQ, LQ or LMQ and sacrificed at day 28. The percentage of CD4 T-cells expressing IFNγ, IL2 or IFNγ/IL2 was determined by ICS and FACS. As a negative control, mice were immunized with excipient (PBS). As a positive control, mice were immunized with SWE adjuvanted spike protein. Data are presented as the mean of n = 4 mice per group.

**Figure 10 pharmaceutics-15-02759-f010:**
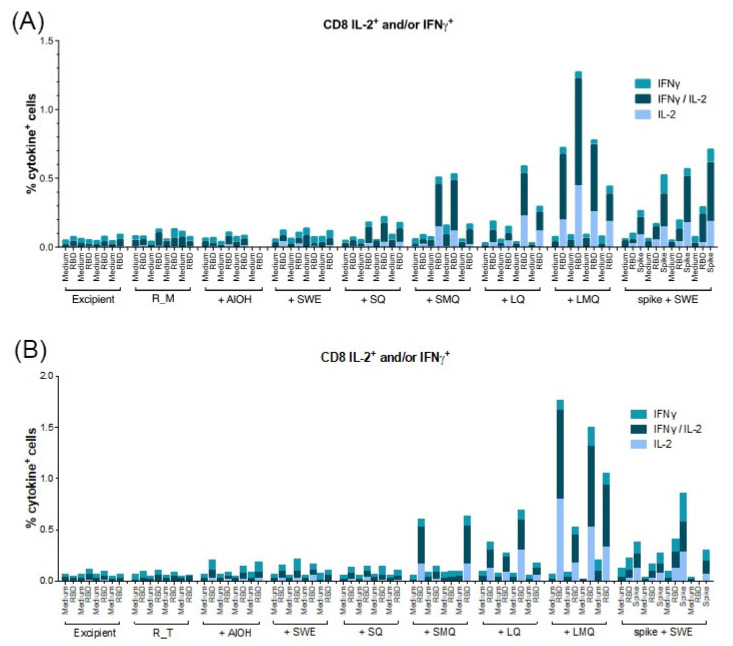
Quantification of CD8 T-cells expressing type 1 cytokines elicited by monomeric RBD (R_M) (top) and trimeric RBD (R_T) (bottom panel) in C57BL/6 mice. C57BL/6 mice were immunized IM on Day 0 and Day 21 with R_M (**A**) or R_T (**B**) adjuvanted with AlOH, SWE, SQ, SMQ, LQ or LMQ and sacrificed at day 28. The percentage of CD8 T-cells expressing IFNγ, IL2 or IFNγ/IL2 was determined by ICS and FACS. As a negative control, mice were immunized with excipient (PBS). As a positive control, mice were immunized with SWE adjuvanted spike protein. Data are presented as the mean of n = 4 mice per group.

**Figure 11 pharmaceutics-15-02759-f011:**
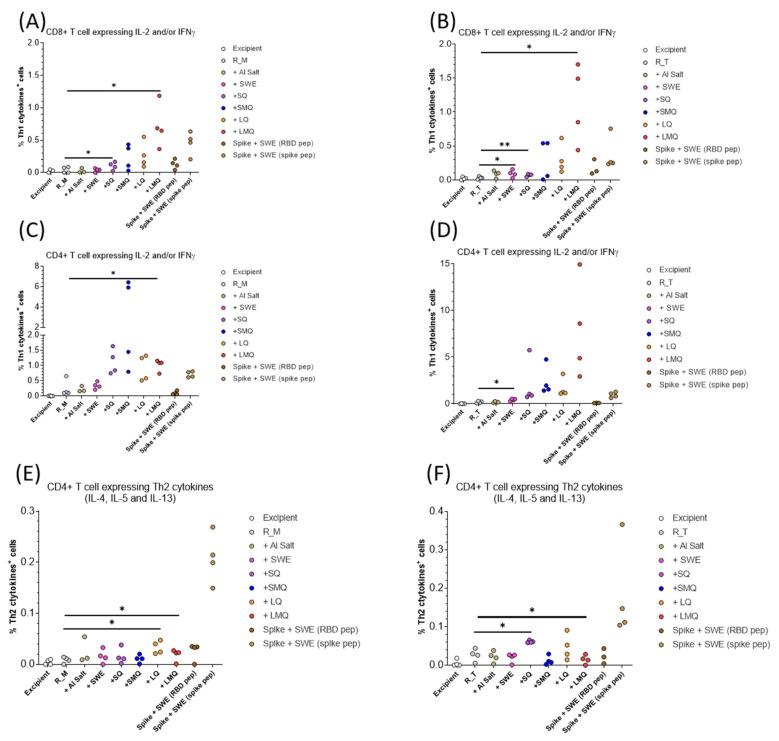
Th1 and Th2 responses in C57BL/6 mice. (**A**–**F**) Quantification of T-cells expressing Th1 cytokines IL-2 and/or IFNγ. (**A**,**B**) CD8+ T-cells. (**C**,**D**) CD4+ T-cells (**E**,**F**) Quantification of CD4+ T-cell expressing Th2 cytokines (IL-4, IL-5 and IL-13). Data are presented as individual n = 4 mice per group. Statistical significance was determined by one-way ANOVA with Dunnett’s multiple comparison (* *p* ≤ 0.05, ** *p* ≤ 0.01).

**Figure 12 pharmaceutics-15-02759-f012:**
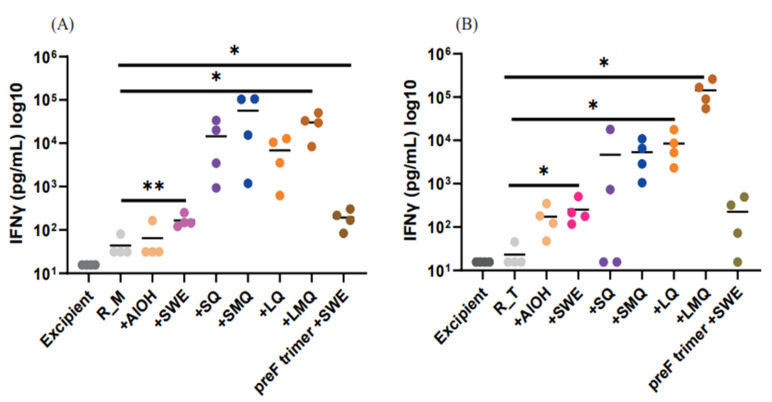
IFNγ production in splenocyte culture supernatant from mice vaccinated with (**A**) R_M and (**B**) R_T formulations containing different adjuvants.C57BL/6 mice were immunized IM on day 0 and Day 21 with R_M (**A**) or R_T (**B**) adjuvanted with AlOH, SWE, SQ, SMQ, LQ or LMQ and sacrificed at day 28. Stimulated splenocytes were measured by ELISA for cells expressing the Type I cytokines IFNγ. As a negative control, mice were immunized with excipient (PBS). As a positive control, mice were immunized with SWE adjuvanted spike protein. Data are presented as individual n = 4 mice per group. Statistical significance was determined by one-way ANOVA with Dunnett’s multiple comparison (* *p* ≤ 0.05, ** *p* ≤ 0.01).

## Data Availability

Relevant data is contained within the article Figures. Any additional data or information can be obtained from the corresponding authors.

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
