# Peer review of "Enhancing Immunogenicity of a Thermostable, Efficacious SARS-CoV-2 Vaccine Formulation through Oligomerization and Adjuvant Choice"

_pharmaceutics, 2023, doi:10.3390/pharmaceutics15122759_

Round 1
Reviewer 1 Report
Comments and Suggestions for Authors
The authors elegantly and carefully examined the influences of oligomerization, adjuvant effects, and temperature stresses of the spike protein of SARS-CoV-2 upon both retained antigenicity and immunogenicity. This work is of prime interest for general vaccine formulations as it will allow maximizing storage conditions, stability and retained immunogenicity of protein-based vaccines.
The work explored various adjuvants and identified liposomal encapsulation (L) in the presence of a TLR4 ligand (M) together with squalene 21 (Q), overall illustrated as a trimeric RBD-LMQ as the ideal formulation for the intended long storage at elevated temperature. The conclusions are adequately defined and firmly supported. This work will be of great value for the community.
This manuscript is highly recommended with only minor revisions as follows"
The abbreviations used throughout are not easy to follow and should be avoided in the abstract to ease overall comprehension and appreciation. Volumes (ml vs mL) are not consistent throughout, particularly in the early section and should be homogenized. Also, R_M and R_T does not seem to be consistent with the RBD_M and RBD_T. The authors should avoid potential confusion in this regard.
Reviewer 2 Report
Comments and Suggestions for Authors
The proposed costimulatory activity of the QS-21 imine binding was through CD2 not the T cell receptor which is generally taken to mean the antigen binding receptor. It would help the reader to be more explanatory.
Give the vaccination site more precisely-thigh?
Describe the mouse corona virus challenge procedure in methods.
What are midpoint titers and why weren't endpoint titers used ?
Many readers will need to know that IgG2c is the C57 equivalent of IgG2a and accordingly a Th1 marker.
The Th2 cytokine response produced by Immunisation with alum salt was low while the IgG1 antibody response was, as expected, high. This would suggest the ability to detect these cytokines was low or the culture system wasn't optimised for these responses. Comment on this with respect to the poor Th2 cytokine responses reported for the other adjuvants.
